# Learning Unknown Markov Decision Processes:
# A Thompson Sampling Approach

**Yi Ouyang**
University of California, Berkeley
ouyangyi@berkeley.edu

**Mukul Gagrani**
University of Southern California
mgagrani@usc.edu

**Ashutosh Nayyar**
University of Southern California
ashutosn@usc.edu

**Rahul Jain**
University of Southern California
rahul.jain@usc.edu

## Abstract

We consider the problem of learning an unknown Markov Decision Process (MDP) that is weakly communicating in the infinite horizon setting. We propose a Thompson Sampling-based reinforcement learning algorithm with dynamic episodes (TSDE). At the beginning of each episode, the algorithm generates a sample from the posterior distribution over the unknown model parameters. It then follows the optimal stationary policy for the sampled model for the rest of the episode. The duration of each episode is dynamically determined by two stopping criteria. The first stopping criterion controls the growth rate of episode length. The second stopping criterion happens when the number of visits to any state-action pair is doubled. We establish $\tilde{O}(HS\sqrt{AT})$ bounds on expected regret under a Bayesian setting, where $S$ and $A$ are the sizes of the state and action spaces, $T$ is time, and $H$ is the bound of the span. This regret bound matches the best available bound for weakly communicating MDPs. Numerical results show it to perform better than existing algorithms for infinite horizon MDPs.

## 1 Introduction

We consider the problem of reinforcement learning by an agent interacting with an environment while trying to minimize the total cost accumulated over time. The environment is modeled by an infinite horizon Markov Decision Process (MDP) with finite state and action spaces. When the environment is perfectly known, the agent can determine optimal actions by solving a dynamic program for the MDP [1]. In reinforcement learning, however, the agent is uncertain about the true dynamics of the MDP. A naive approach to an unknown model is the *certainty equivalence principle*. The idea is to estimate the unknown MDP parameters from available information and then choose actions as if the estimates are the true parameters. But it is well-known in adaptive control theory that the certainty equivalence principle may lead to suboptimal performance due to the lack of exploration [2]. This issue actually comes from the fundamental exploitation-exploration trade-off: the agent wants to exploit available information to minimize cost, but it also needs to explore the environment to learn system dynamics.

One common way to handle the exploitation-exploration trade-off is to use the *optimism in the face of uncertainty* (OFU) principle [3]. Under this principle, the agent constructs confidence sets for the system parameters at each time, find the optimistic parameters that are associated with the minimum cost, and then selects an action based on the optimistic parameters. The optimism procedure encourages exploration for rarely visited states and actions. Several optimistic algorithms are proved to possess strong theoretical performance guarantees [4–10].

An alternative way to incentivize exploration is the Thompson Sampling (TS) or Posterior Sampling method. The idea of TS was first proposed by Thompson in [11] for stochastic bandit problems. It has been applied to MDP environments [12–17] where the agent computes the posterior distribution of unknown parameters using observed information and a prior distribution. A TS algorithm generally proceeds in episodes: at the beginning of each episode a set of MDP parameters is randomly sampled from the posterior distribution, then actions are selected based on the sampled model during the episode. TS algorithms have the following advantages over optimistic algorithms. First, TS algorithms can easily incorporate problem structures through the prior distribution. Second, they are more computationally efficient since a TS algorithm only needs to solve the sampled MDP, while an optimistic algorithm requires solving all MDPs that lie within the confident sets. Third, empirical studies suggest that TS algorithms outperform optimistic algorithms in bandit problems [18, 19] as well as in MDP environments [13, 16, 17].

Due to the above advantages, we focus on TS algorithms for the MDP learning problem. The main challenge in the design of a TS algorithm is the lengths of the episodes. For finite horizon MDPs under the episodic setting, the length of each episode can be set as the time horizon [13]. When there exists a recurrent state under any stationary policy, the TS algorithm of [15] starts a new episode whenever the system enters the recurrent state. However, the above methods to end an episode can not be applied to MDPs without the special features. The work of [16] proposed a dynamic episode schedule based on the doubling trick used in [7], but a mistake in their proof of regret bound was pointed out by [20]. In view of the mistake in [16], there is no TS algorithm with strong performance guarantees for general MDPs to the best of our knowledge.

We consider the most general subclass of weakly communicating MDPs in which meaningful finite time regret guarantees can be analyzed. We propose the Thompson Sampling with Dynamic Episodes (TSDE) learning algorithm. In TSDE, there are two stopping criteria for an episode to end. The first stopping criterion controls the growth rate of episode length. The second stopping criterion is the doubling trick similar to the one in [7–10, 16] that stops when the number of visits to any state-action pair is doubled. Under a Bayesian framework, we show that the expected regret of TSDE accumulated up to time $T$ is bounded by $\tilde{O}(HS\sqrt{AT})$ where $\tilde{O}$ hides logarithmic factors. Here $S$ and $A$ are the sizes of the state and action spaces, $T$ is time, and $H$ is the bound of the span. This regret bound matches the best available bound for weakly communicating MDPs [7], and it matches the theoretical lower bound in order of $T$ except for logarithmic factors. We present numerical results that show that TSDE actually outperforms current algorithms with known regret bounds that have the same order in $T$ for a benchmark MDP problem as well as randomly generated MDPs.

## 2 Problem Formulation

### 2.1 Preliminaries

An infinite horizon Markov Decision Process (MDP) is described by $(\mathcal{S}, \mathcal{A}, c, \theta)$. Here $\mathcal{S}$ is the state space, $\mathcal{A}$ is the action space, $c : \mathcal{S} \times \mathcal{A} \rightarrow [0, 1]^1$ is the cost function, and $\theta : \mathcal{S}^2 \times \mathcal{A} \rightarrow [0, 1]$ represents the transition probabilities such that $\theta(s'|s, a) = \mathbb{P}(s_{t+1} = s'|s_t = s, a_t = a)$ where $s_t \in \mathcal{S}$ and $a_t \in \mathcal{A}$ are the state and the action at $t = 1, 2, 3 \ldots$. We assume that $\mathcal{S}$ and $\mathcal{A}$ are finite spaces with sizes $S \geq 2$ and $A \geq 2$, and the initial state $s_1$ is a known and fixed state. A stationary policy is a deterministic map $\pi : \mathcal{S} \rightarrow \mathcal{A}$ that maps a state to an action. The average cost per stage of a stationary policy is defined as

$$J_\pi(\theta) = \limsup_{T \to \infty} \frac{1}{T} \mathbb{E}\Big[ \sum_{t=1}^{T} c(s_t, a_t) \Big].$$

Here we use $J_\pi(\theta)$ to explicitly show the dependency of the average cost on $\theta$.

To have meaningful finite time regret bounds, we consider the subclass of weakly communicating MDPs defined as follows.

**Definition 1.** *An MDP is weakly communicating (or weak accessible) if its states can be partitioned into two subsets: in the first subset all states are transient under every stationary policy, and every two states in the second subset can be reached from each other under some stationary policy.*

From MDP theory [1], we know that if the MDP is weakly communicating, the optimal average cost per stage $J(\theta) = \min_\pi J_\pi(\theta)$ satisfies the Bellman equation

$$J(\theta) + v(s, \theta) = \min_{a \in \mathcal{A}} \left\{ c(s, a) + \sum_{s' \in \mathcal{S}} \theta(s'|s, a) v(s', \theta) \right\} \tag{1}$$

for all $s \in \mathcal{S}$. The corresponding optimal stationary policy $\pi^*$ is the minimizer of the above optimization given by

$$a = \pi^*(s, \theta). \tag{2}$$

Since the cost function $c(s, a) \in [0, 1]$, $J(\theta) \in [0, 1]$ for all $\theta$. If $v$ satisfies the Bellman equation, $v$ plus any constant also satisfies the Bellman equation. Without loss of generality, let $\min_{s \in \mathcal{S}} v(s, \theta) = 0$ and define the span of the MDP as $sp(\theta) = \max_{s \in \mathcal{S}} v(s, \theta)$. [2]

We define $\Omega_*$ to be the set of all $\theta$ such that the MDP with transition probabilities $\theta$ is weakly communicating, and there exists a number $H$ such that $sp(\theta) \leq H$. We will focus on MDPs with transition probabilities in the set $\Omega_*$.

## 2.2 Reinforcement Learning for Weakly Communicating MDPs

We consider the reinforcement learning problem of an agent interacting with a random weakly communicating MDP $(\mathcal{S}, \mathcal{A}, c, \theta_*)$. We assume that $\mathcal{S}$, $\mathcal{A}$ and the cost function $c$ are completely known to the agent. The actual transition probabilities $\theta_*$ is randomly generated at the beginning before the MDP interacts with the agent. The value of $\theta_*$ is then fixed but unknown to the agent. The complete knowledge of the cost is typical as in [7, 15]. Algorithms can generally be extended to the unknown costs/rewards case at the expense of some constant factor for the regret bound.

At each time $t$, the agent selects an action according to $a_t = \phi_t(h_t)$ where $h_t = (s_1, s_2, \ldots, s_t, a_1, a_2, \ldots, a_{t-1})$ is the history of states and actions. The collection $\phi = (\phi_1, \phi_2 \ldots)$ is called a learning algorithm. The functions $\phi_t$ allow for the possibility of randomization over actions at each time.

We focus on a Bayesian framework for the unknown parameter $\theta_*$. Let $\mu_1$ be the prior distribution for $\theta_*$, i.e., for any set $\Theta$, $\mathbb{P}(\theta_* \in \Theta) = \mu_1(\Theta)$. We make the following assumptions on $\mu_1$.

**Assumption 1.** *The support of the prior distribution $\mu_1$ is a subset of $\Omega_*$. That is, the MDP is weakly communicating and $sp(\theta_*) \leq H$.*

In this Bayesian framework, we define the expected regret (also called Bayesian regret or Bayes risk) of a learning algorithm $\phi$ up to time $T$ as

$$R(T, \phi) = \mathbb{E} \left[ \sum_{t=1}^{T} \left[ c(s_t, a_t) - J(\theta_*) \right] \right] \tag{3}$$

where $s_t, a_t, t = 1, \ldots, T$ are generated by $\phi$ and $J(\theta_*)$ is the optimal per stage cost of the MDP. The above expectation is with respect to the prior distribution $\mu_1$ for $\theta_*$, the randomness in state transitions, and the randomized algorithm. The expected regret is an important metric to quantify the performance of a learning algorithm.

## 3 Thompson Sampling with Dynamic Episodes

In this section, we propose the Thompson Sampling with Dynamic Episodes (TSDE) learning algorithm. The input of TSDE is the prior distribution $\mu_1$. At each time $t$, given the history $h_t$, the agent can compute the posterior distribution $\mu_t$ given by $\mu_t(\Theta) = \mathbb{P}(\theta_* \in \Theta | h_t)$ for any set $\Theta$. Upon applying the action $a_t$ and observing the new state $s_{t+1}$, the posterior distribution at $t + 1$ can be updated according to Bayes' rule as

$$\mu_{t+1}(d\theta) = \frac{\theta(s_{t+1}|s_t, a_t) \mu_t(d\theta)}{\int \theta'(s_{t+1}|s_t, a_t) \mu_t(d\theta')}. \tag{4}$$

Let $N_t(s,a)$ be the number of visits to any state-action pair $(s,a)$ before time $t$. That is,

$$N_t(s,a) = |\{\tau < t : (s_\tau, a_\tau) = (s,a)\}|. \tag{5}$$

With these notations, TSDE is described as follows.

---

**Algorithm 1** Thompson Sampling with Dynamic Episodes (TSDE)

---

Input: $\mu_1$
Initialization: $t \leftarrow 1$, $t_k \leftarrow 0$
**for** episodes $k = 1, 2, ...$ **do**
    $T_{k-1} \leftarrow t - t_k$
    $t_k \leftarrow t$
    Generate $\theta_k \sim \mu_{t_k}$ and compute $\pi_k(\cdot) = \pi^*(\cdot, \theta_k)$ from (1)-(2)
    **while** $t \leq t_k + T_{k-1}$ and $N_t(s,a) \leq 2N_{t_k}(s,a)$ for all $(s,a) \in \mathcal{S} \times \mathcal{A}$ **do**
        Apply action $a_t = \pi_k(s_t)$
        Observe new state $s_{t+1}$
        Update $\mu_{t+1}$ according to (4)
        $t \leftarrow t + 1$
    **end while**
**end for**

---

The TSDE algorithm operates in episodes. Let $t_k$ be start time of the $k$th episode and $T_k = t_{k+1} - t_k$ be the length of the episode with the convention $T_0 = 1$. From the description of the algorithm, $t_1 = 1$ and $t_{k+1}, k \geq 1$, is given by

$$t_{k+1} = \min\{t > t_k : \quad t > t_k + T_{k-1} \text{ or } N_t(s,a) > 2N_{t_k}(s,a) \text{ for some } (s,a)\}. \tag{6}$$

At the beginning of episode $k$, a parameter $\theta_k$ is sampled from the posterior distribution $\mu_{t_k}$. During each episode $k$, actions are generated from the optimal stationary policy $\pi_k$ for the sampled parameter $\theta_k$. One important feature of TSDE is that its episode lengths are not fixed. The length $T_k$ of each episode is dynamically determined according to two stopping criteria: (i) $t > t_k + T_{k-1}$, and (ii) $N_t(s,a) > 2N_{t_k}(s,a)$ for some state-action pair $(s,a)$. The first stopping criterion provides that the episode length grows at a linear rate without triggering the second criterion. The second stopping criterion ensures that the number of visits to any state-action pair $(s,a)$ during an episode should not be more than the number visits to the pair before this episode.

**Remark 1.** *Note that TSDE only requires the knowledge of $\mathcal{S}$, $\mathcal{A}$, c, and the prior distribution $\mu_1$. TSDE can operate without the knowledge of time horizon $T$, the bound $H$ on span used in [7], and any knowledge about the actual $\theta_*$ such as the recurrent state needed in [15].*

### 3.1 Main Result

**Theorem 1.** *Under Assumption 1,*

$$R(T, \text{TSDE}) \leq (H+1)\sqrt{2SAT\log(T)} + 49HS\sqrt{AT\log(AT)}.$$

The proof of Theorem 1 appears in Section 4.

**Remark 2.** *Note that our regret bound has the same order in $H, S, A$ and $T$ as the optimistic algorithm in [7] which is the best available bound for weakly communicating MDPs. Moreover, the bound does not depend on the prior distribution or other problem-dependent parameters such as the recurrent time of the optimal policy used in the regret bound of [15].*

### 3.2 Approximation Error

At the beginning of each episode, TSDE computes the optimal stationary policy $\pi_k$ for the parameter $\theta_k$. This step requires the solution to a fixed finite MDP. Policy iteration or value iteration can be used to solve the sampled MDP, but the resulting stationary policy may be only approximately optimal in practice. We call $\pi$ an $\epsilon-$approximate policy if

$$c(s, \pi(s)) + \sum_{s' \in \mathcal{S}} \theta(s'|s, \pi(s))v(s', \theta) \leq \min_{a \in \mathcal{A}} \left\{ c(s,a) + \sum_{s' \in \mathcal{S}} \theta(s'|s,a)v(s', \theta) \right\} + \epsilon.$$

When the algorithm returns an $\epsilon_k-$approximate policy $\tilde{\pi}_k$ instead of the optimal stationary policy $\pi_k$ at episode $k$, we have the following regret bound in the presence of such approximation error.

**Theorem 2.** *If TSDE computes an $\epsilon_k-$approximate policy $\tilde{\pi}_k$ instead of the optimal stationary policy $\pi_k$ at each episode $k$, the expected regret of TSDE satisfies*

$$R(T, \text{TSDE}) \leq \tilde{O}(HS\sqrt{AT}) + \mathbb{E}\Big[ \sum_{k:t_k \leq T} T_k \epsilon_k \Big].$$

*Furthermore, if $\epsilon_k \leq \frac{1}{k+1}$, $\mathbb{E}\Big[ \sum_{k:t_k \leq T} T_k \epsilon_k \Big] \leq \sqrt{2SAT \log(T)}.$*

Theorem 2 shows that the approximation error in the computation of optimal stationary policy is only additive to the regret under TSDE. The regret bound would remain $\tilde{O}(HS\sqrt{AT})$ if the approximation error is such that $\epsilon_k \leq \frac{1}{k+1}$. The proof of Theorem 2 is in the appendix due to the lack of space.

## 4  Analysis

### 4.1  Number of Episodes

To analyze the performance of TSDE over $T$ time steps, define $K_T = \arg\max\{k : t_k \leq T\}$ be the number of episodes of TSDE until time $T$. Note that $K_T$ is a random variable because the number of visits $N_t(x, u)$ depends on the dynamical state trajectory. In the analysis for time $T$ we use the convention that $t_{(K_T+1)} = T + 1$. We provide an upper bound on $K_T$ as follows.

**Lemma 1.**

$$K_T \leq \sqrt{2SAT \log(T)}.$$

*Proof.* Define macro episodes with start times $t_{n_i}, i = 1, 2, \ldots$ where $t_{n_1} = t_1$ and

$$t_{n_{i+1}} = \min\{t_k > t_{n_i} : \quad N_{t_k}(s, a) > 2N_{t_{k-1}}(s, a) \text{ for some } (s, a)\}.$$

The idea is that each macro episode starts when the second stopping criterion happens. Let $M$ be the number of macro episodes until time $T$ and define $n_{(M+1)} = K_T + 1$.

Let $\tilde{T}_i = \sum_{k=n_i}^{n_{i+1}-1} T_k$ be the length of the $i$th macro episode. By the definition of macro episodes, any episode except the last one in a macro episode must be triggered by the first stopping criterion. Therefore, within the $i$th macro episode, $T_k = T_{k-1} + 1$ for all $k = n_i, n_i + 1, \ldots, n_{i+1} - 2$. Hence,

$$\tilde{T}_i = \sum_{k=n_i}^{n_{i+1}-1} T_k = \sum_{j=1}^{n_{i+1}-n_i-1} (T_{n_i-1} + j) + T_{n_{i+1}-1}$$

$$\geq \sum_{j=1}^{n_{i+1}-n_i-1} (j+1) + 1 = 0.5(n_{i+1} - n_i)(n_{i+1} - n_i + 1).$$

Consequently, $n_{i+1} - n_i \leq \sqrt{2\tilde{T}_i}$ for all $i = 1, \ldots, M$. From this property we obtain

$$K_T = n_{M+1} - 1 = \sum_{i=1}^{M}(n_{i+1} - n_i) \leq \sum_{i=1}^{M}\sqrt{2\tilde{T}_i}. \tag{7}$$

Using (7) and the fact that $\sum_{i=1}^{M} \tilde{T}_i = T$ we get

$$K_T \leq \sum_{i=1}^{M}\sqrt{2\tilde{T}_i} \leq \sqrt{M\sum_{i=1}^{M} 2\tilde{T}_i} = \sqrt{2MT} \tag{8}$$

where the second inequality is Cauchy-Schwarz.

From Lemma 6 in the appendix, the number of macro episodes $M \leq SA \log(T)$. Substituting this bound into (8) we obtain the result of this lemma. $\quad\square$

**Remark 3.** *TSDE computes the optimal stationary policy of a finite MDP at each episode. Lemma 1 ensures that such computation only needs to be done at a sublinear rate of $\sqrt{2SAT \log(T)}$.*

## 4.2 Regret Bound

As discussed in [13, 20, 21], one key property of Thompson/Posterior Sampling algorithms is that for any function $f$, $\mathbb{E}[f(\theta_t)] = \mathbb{E}[f(\theta_*)]$ if $\theta_t$ is sampled from the posterior distribution at time $t$. This property leads to regret bounds for algorithms with fixed sampling episodes since the start time $t_k$ of each episode is deterministic. However, our TSDE algorithm has dynamic episodes that requires us to have the stopping-time version of the above property.

**Lemma 2.** *Under TSDE, $t_k$ is a stopping time for any episode $k$. Then for any measurable function $f$ and any $\sigma(h_{t_k})-$measurable random variable $X$, we have*

$$\mathbb{E}\left[f(\theta_k, X)\right] = \mathbb{E}\left[f(\theta_*, X)\right]$$

*Proof.* From the definition (6), the start time $t_k$ is a stopping-time, i.e. $t_k$ is $\sigma(h_{t_k})-$measurable. Note that $\theta_k$ is randomly sampled from the posterior distribution $\mu_{t_k}$. Since $t_k$ is a stopping time, $t_k$ and $\mu_{t_k}$ are both measurable with respect to $\sigma(h_{t_k})$. From the assumption, $X$ is also measurable with respect to $\sigma(h_{t_k})$. Then conditioned on $h_{t_k}$, the only randomness in $f(\theta_k, X)$ is the random sampling in the algorithm. This gives the following equation:

$$\mathbb{E}\left[f(\theta_k, X)|h_{t_k}\right] = \mathbb{E}\left[f(\theta_k, X)|h_{t_k}, t_k, \mu_{t_k}\right] = \int f(\theta, X)\mu_{t_k}(d\theta) = \mathbb{E}\left[f(\theta_*, X)|h_{t_k}\right] \quad (9)$$

since $\mu_{t_k}$ is the posterior distribution of $\theta_*$ given $h_{t_k}$. Now the result follows by taking the expectation of both sides. $\qquad\square$

For $t_k \leq t < t_{k+1}$ in episode $k$, the Bellman equation (1) holds by Assumption 1 for $s = s_t$, $\theta = \theta_k$ and action $a_t = \pi_k(s_t)$. Then we obtain

$$c(s_t, a_t) = J(\theta_k) + v(s_t, \theta_k) - \sum_{s' \in \mathcal{S}} \theta_k(s'|s_t, a_t)v(s', \theta_k). \quad (10)$$

Using (10), the expected regret of TSDE is equal to

$$\mathbb{E}\left[\sum_{k=1}^{K_T} \sum_{t=t_k}^{t_{k+1}-1} c(s_t, a_t)\right] - T\,\mathbb{E}\left[J(\theta_*)\right]$$

$$= \mathbb{E}\left[\sum_{k=1}^{K_T} T_k J(\theta_k)\right] - T\,\mathbb{E}\left[J(\theta_*)\right] + \mathbb{E}\left[\sum_{k=1}^{K_T} \sum_{t=t_k}^{t_{k+1}-1} \left[v(s_t, \theta_k) - \sum_{s' \in \mathcal{S}} \theta_k(s'|s_t, a_t)v(s', \theta_k)\right]\right]$$

$$= R_0 + R_1 + R_2, \quad (11)$$

where $R_0$, $R_1$ and $R_2$ are given by

$$R_0 = \mathbb{E}\left[\sum_{k=1}^{K_T} T_k J(\theta_k)\right] - T\,\mathbb{E}\left[J(\theta_*)\right],$$

$$R_1 = \mathbb{E}\left[\sum_{k=1}^{K_T} \sum_{t=t_k}^{t_{k+1}-1} \left[v(s_t, \theta_k) - v(s_{t+1}, \theta_k)\right]\right],$$

$$R_2 = \mathbb{E}\left[\sum_{k=1}^{K_T} \sum_{t=t_k}^{t_{k+1}-1} \left[v(s_{t+1}, \theta_k) - \sum_{s' \in \mathcal{S}} \theta_k(s'|s_t, a_t)v(s', \theta_k)\right]\right].$$

We proceed to derive bounds on $R_0$, $R_1$ and $R_2$.

Based on the key property of Lemma 2, we derive an upper bound on $R_0$.

**Lemma 3.** *The first term $R_0$ is bounded as*

$$R_0 \leq \mathbb{E}[K_T].$$

*Proof.* From monotone convergence theorem we have

$$R_0 = \mathbb{E}\Big[\sum_{k=1}^{\infty} \mathbb{1}_{\{t_k \leq T\}} T_k J(\theta_k)\Big] - T \mathbb{E}\Big[J(\theta_*)\Big] = \sum_{k=1}^{\infty} \mathbb{E}\Big[\mathbb{1}_{\{t_k \leq T\}} T_k J(\theta_k)\Big] - T \mathbb{E}\Big[J(\theta_*)\Big].$$

Note that the first stopping criterion of TSDE ensures that $T_k \leq T_{k-1} + 1$ for all $k$. Because $J(\theta_k) \geq 0$, each term in the first summation satisfies

$$\mathbb{E}\Big[\mathbb{1}_{\{t_k \leq T\}} T_k J(\theta_k)\Big] \leq \mathbb{E}\Big[\mathbb{1}_{\{t_k \leq T\}}(T_{k-1} + 1) J(\theta_k)\Big].$$

Note that $\mathbb{1}_{\{t_k \leq T\}}(T_{k-1} + 1)$ is measurable with respect to $\sigma(h_{t_k})$. Then, Lemma 2 gives

$$\mathbb{E}\Big[\mathbb{1}_{\{t_k \leq T\}}(T_{k-1} + 1) J(\theta_k)\Big] = \mathbb{E}\Big[\mathbb{1}_{\{t_k \leq T\}}(T_{k-1} + 1) J(\theta_*)\Big].$$

Combining the above equations we get

$$R_0 \leq \sum_{k=1}^{\infty} \mathbb{E}\Big[\mathbb{1}_{\{t_k \leq T\}}(T_{k-1} + 1) J(\theta_*)\Big] - T \mathbb{E}\Big[J(\theta_*)\Big]$$

$$= \mathbb{E}\Big[\sum_{k=1}^{K_T}(T_{k-1} + 1) J(\theta_*)\Big] - T \mathbb{E}\Big[J(\theta_*)\Big]$$

$$= \mathbb{E}\Big[K_T J(\theta_*)\Big] + \mathbb{E}\Big[\Big(\sum_{k=1}^{K_T} T_{k-1} - T\Big) J(\theta_*)\Big] \leq \mathbb{E}\Big[K_T\Big]$$

where the last equality holds because $J(\theta_*) \leq 1$ and $\sum_{k=1}^{K_T} T_{k-1} = T_0 + \sum_{k=1}^{K_T - 1} T_k \leq T$. □

Note that the first stopping criterion of TSDE plays a crucial role in the proof of Lemma 3. It allows us to bound the length of an episode using the length of the previous episode which is measurable with respect to the information at the beginning of the episode.

The other two terms $R_1$ and $R_2$ of the regret are bounded in the following lemmas. Their proofs follow similar steps to those in [13, 16]. The proofs are in the appendix due to the lack of space.

**Lemma 4.** *The second term $R_1$ is bounded as*

$$R_1 \leq \mathbb{E}[HK_T].$$

**Lemma 5.** *The third term $R_2$ is bounded as*

$$R_2 \leq 49HS\sqrt{AT \log(AT)}.$$

We are now ready to prove Theorem 1.

*Proof of Theorem 1.* From (11), $R(T, \text{TSDE}) = R_0 + R_1 + R_2 \leq \mathbb{E}[K_T] + \mathbb{E}[HK_T] + R_2$ where the inequality comes from Lemma 3, Lemma 4. Then the claim of the theorem directly follows from Lemma 1 and Lemma 5. □

## 5 Simulations

In this section, we compare through simulations the performance of TSDE with three learning algorithms with the same regret order: UCRL2 [8], TSMDP [15], and Lazy PSRL [16]. UCRL2 is an optimistic algorithm with similar regret bounds. TSMDP and Lazy PSRL are TS algorithms for infinite horizon MDPs. TSMDP has the same regret order in $T$ given a recurrent state for resampling. The original regret analysis for Lazy PSRL is incorrect, but the regret bounds are conjectured to be correct [20]. We chose $\delta = 0.05$ for the implementation of UCRL2 and assume an independent Dirichlet prior with parameters $[0.1, \ldots, 0.1]$ over the transition probabilities for all TS algorithms.

We consider two environments: randomly generated MDPs and the RiverSwim example [22]. For randomly generated MDPs, we use the independent Dirichlet prior over 6 states and 2 actions but

with a fixed cost. We select the resampling state $s_0 = 1$ for TSMDP here since all states are recurrent under the Dirichlet prior. The RiverSwim example models an agent swimming in a river who can choose to swim either left or right. The MDP consists of six states arranged in a chain with the agent starting in the leftmost state ($s = 1$). If the agent decides to move left i.e with the river current then he is always successful but if he decides to move right he might fail with some probability. The cost function is given by: $c(s, a) = 0.8$ if $s = 1$, $a =$ left; $c(s, a) = 0$ if $s = 6$, $a =$ right; and $c(s, a) = 1$ otherwise. The optimal policy is to swim right to reach the rightmost state which minimizes the cost. For TSMDP in RiverSwim, we consider two versions with $s_0 = 1$ and with $s_0 = 3$ for the resampling state. We simulate 500 Monte Carlo runs for both the examples and run for $T = 10^5$.

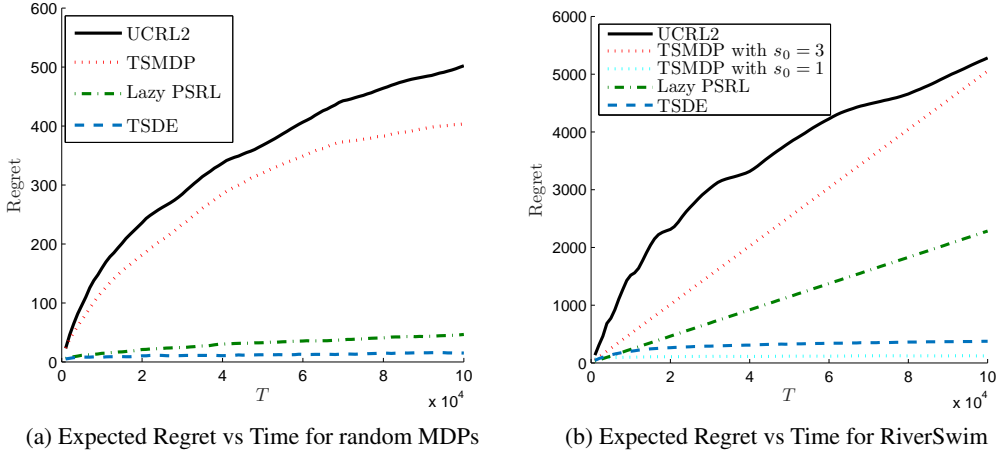

(a) Expected Regret vs Time for random MDPs        (b) Expected Regret vs Time for RiverSwim

Figure 1: Simulation Results

From Figure 1(a) we can see that TSDE outperforms all the three algorithms in randomly generated MDPs. In particular, there is a significant gap between the regret of TSDE and that of UCRL2 and TSMDP. The poor performance of UCRL2 assures the motivation to consider TS algorithms. From the specification of TSMDP, its performance heavily hinges on the choice of an appropriate resampling state which is not possible for a general unknown MDP. This is reflected in the randomly generated MDPs experiment.

In the RiverSwim example, Figure 1(b) shows that TSDE significantly outperforms UCRL2, Lazy PSRL, and TSMDP with $s_0 = 3$. Although TSMDP with $s_0 = 1$ performs slightly better than TSDE, there is no way to pick this specific $s_0$ if the MDP is unknown in practice. Since Lazy PSRL is also equipped with the doubling trick criterion, the performance gap between TSDE and Lazy PSRL highlights the importance of the first stopping criterion on the growth rate of episode length. We also like to point out that in this example, the MDP is fixed and is not generated from the Dirichlet prior. Therefore, we conjecture that TSDE also has the same regret bounds under a non-Bayesian setting.

## 6   Conclusion

We propose the Thompson Sampling with Dynamic Episodes (TSDE) learning algorithm and establish $\tilde{O}(HS\sqrt{AT})$ bounds on expected regret for the general subclass of weakly communicating MDPs. Our result fills a gap in the theoretical analysis of Thompson Sampling for MDPs. Numerical results validate that the TSDE algorithm outperforms other learning algorithms for infinite horizon MDPs.

The TSDE algorithm determines the end of an episode by two stopping criteria. The second criterion comes from the doubling trick used in many reinforcement learning algorithms. But the first criterion on the linear growth rate of episode length seems to be a new idea for episodic learning algorithms. The stopping criterion is crucial in the proof of regret bound (Lemma 3). The simulation results of TSDE versus Lazy PSRL further shows that this criterion is not only a technical constraint for proofs, it indeed helps balance exploitation and exploration.

**Acknowledgments**

Yi Ouyang would like to thank Yang Liu from Harvard University for helpful discussions. Rahul Jain and Ashutosh Nayyar were supported by NSF Grants 1611574 and 1446901.

## Footnotes

[1]Since $\mathcal{S}$ and $\mathcal{A}$ are finite, we can normalize the cost function to $[0, 1]$ without loss of generality.

[2]See [7]for a discussion on the connection of the span with other parameters such as the diameter appearing in the lower bound on regret.

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
