[Reviews · NeurIPS 2017]

Reviewer 1



The paper addresses the exploration/exploitation problem of MDPs, and offers theoretical bounds on regret in infinite-horizon weakly communicating MDPs when using Thompson Sampling to generate policies with carefully controlled trajectory lengths. The bounds are asymptotically the same as the best bound for regret with any approach on weakly communicating MDPs. There are some experiments showing competitive or improved performance with other approaches. The paper is well-written and clear, though necessarily theoretically dense. I also did not find any errors. In terms of impact, it seems to be highly relevant to a small audience, though as an RL person who does not regularly think about regret, this is my least confident part of the review. The experiments are small and unambitious, but appropriate for the theoretical nature of the paper.

Reviewer 2



The authors present a reinforcement learning algorithm based on posterior sampling and show that it achieves an expected Bayesian regret bound that matches the best existing bound for weakly communicating MDPs. The paper is easy to read and seems technically correct, although I did not check all the proofs. My main concern is with respect to the main result, which is the regret bound. The scaling O(HS sqrt{AT}) for the (arguably weaker) expected regret is similar to the best existing bounds for the worst-case total regret. This does not explain why Thompson sampling is superior to other optimistic algorithms -- as illustrated in the empirical examples. In particular, the advantage of the proposed policy-update schedule is unclear.

Reviewer 3



The paper proposes TSDE, a posterior sampling algorithm for RL in the average reward infinite horizon setting. This algorithm uses dynamic episodes but unlike Lazy-PSRL avoids technical issues by not only terminating an episode when an observation count doubled but also terminating episodes when they become too long. This ensures that the episode length cannot grow faster than linear and ultimately a Bayesian regret bound of O(HS(AT)^.5) is shown. Posterior sampling methods typically outperform UCB-type algorithms and therefore a posterior sampling algorithm for non-episodic RL with rigorous regret bounds is desirable. This paper proposes such an algorithm, which is of high interest. The paper is well written and overall easy to follow. I have verified all proofs but Theorem 2 and found no mistakes. The presentation is rigorous and the main insights are discussed sufficiently. This paper is therefore a good contribution to the NIPS community. A comparison against UCRL2 with empirically tightened confidence intervals would have been interesting. An more extended comparison of TSDE for MDPs with increasing size would have also been interesting to empirically determine the dependency of the regret on the problem quantities. This would have given insights on whether we can expect to prove a tighter regret bound (e.g. with \sqrt(S) dependency) without changing the algorithm. Nonetheless, all of this is somewhat out of the scope of this paper. Minor detail: The condition on the bottom of page 4 should hold for all s.